# Transperineal US-MRI Fusion-Guided Biopsy for the Detection of Clinical Significant Prostate Cancer: A Systematic Review and Meta-Analysis Comparing Cognitive and Software-Assisted Technique

**DOI:** 10.3390/cancers15133443

**Published:** 2023-06-30

**Authors:** Giacomo Maria Pirola, Daniele Castellani, Luca Orecchia, Carlo Giulioni, Marilena Gubbiotti, Emanuele Rubilotta, Martina Maggi, Jeremy Yuen-Chun Teoh, Vineet Gauhar, Angelo Naselli

**Affiliations:** 1Department of Urology, San Giuseppe Hospital, IRCCS Multimedica, 20123 Milano, Italy; angelo.naselli@libero.it; 2Department of Urology, University Hospital “Ospedali Riuniti”, Polytechnic University of Marche Region, 60131 Ancona, Italy; castellanidaniele@gmail.com (D.C.); carlo.giulioni9@gmail.com (C.G.); 3Urology Unit, Fondazione PTV Policlinico Tor Vergata University Hospital, 00133 Rome, Italy; luca.orecchia@hotmail.com; 4Department of Urology, Usl Toscana Sud Est, San Donato Hospital, 52100 Arezzo, Italy; marilena.gubbiotti@gmail.com; 5Department of Urology, A.O.U.I. Verona University, 37129 Verona, Italy; emanuele.rubilotta@aovr.veneto.it; 6Maternal Infant and Urologic Sciences, Policlinico Umberto I Hospital, “Sapienza” University of Rome, 00185 Rome, Italy; martina.maggi@uniroma1.it; 7Department of Surgery, S.H. Ho Urology Center, The Chinese University of Hong Kong, Hong Kong, China; jeremyteoh@surgery.cuhk.edu.hk; 8Department of Urology, Ng Teng Fong General Hospital, National University Health System, Singapore 119074, Singapore; vineetgaauhaar@gmail.com

**Keywords:** prostate cancer, prostate biopsy, magnetic resonance imaging, US-MRI fusion biopsy, clinically significant prostate cancer, transperineal prostate biopsy

## Abstract

**Simple Summary:**

US-MRI fusion biopsy is established as a technique of reference for the detection of clinically significative prostate cancer compared to the ultrasound “template” technique. Several software have been developed to aid clinicians to perform a real-time fusion between MRI and US prostate imaging; however, the images can also be mentally superimposed by the operator performing a cognitive fusion. Many papers in the literature describe the feasibility and efficacy of these two techniques, but few have performed a direct comparison between them. Therefore, we selected all comparative studies with the aim to perform a meta-analysis to find if one technique leads to an improvement in the detection rate of clinically significant prostate cancer (csPCa) for biopsies performed with a transperineal approach. Our findings indicate that csPCa detection is comparable between these techniques. Thus, clinicians can choose to perform a cognitive or software-assisted biopsy in accordance with their personal experience or technologic availability without the potential risk of offering an underperforming methodology.

**Abstract:**

Introduction: We aimed to find potential differences in clinically significant prostate cancer (csPCa) detection rates between transperineal software-assisted fusion biopsy (saFB) and cognitive fusion biopsies (cFB). Methods: A systematic review of the literature was performed to identify comparative studies using PubMed, EMBASE, and Scopus according to the PICOS criteria. Cancer detection and complication rates were pooled using the Cochran–Mantel–Haenszel method with the random effect model and reported as odds ratios (ORs), 95% confidence intervals (CI), and *p*-values. A meta-analysis was performed using Review Manager (RevMan) 5.4 software by Cochrane Collaboration. The quality assessment of the included studies was performed using the Cochrane Risk of Bias tool, using RoB 2 for randomized studies and ROBINS-I for retrospective and nonrandomized ones. Results: Eight studies were included for the meta-analysis, including 1149 cases in software-based and 963 cases in cognitive fusion biopsy. The detection rates of csPCa were similar between the two groups (OR 1.01, 95% CI 0.74–1.37, *p* = 0.95). Study heterogeneity was low (I2 55%). Conclusion: There is no actual evidence of the superiority of saFB over cFB in terms of the csPCa detection rate. Operator experience and software availability can drive the choice of one fusion technique over the other.

## 1. Introduction

Prostate cancer (PCa) is nowadays the most common diagnosis of malignant cancer in European men, and overall, in males of more developed countries [1]. PCa diagnosis relies on prostate biopsy (PB), which can either be performed with the transrectal (TR) or transperineal (TP) approach [2]. Even if TP-PB was demonstrated to increase the PCa detection rate compared to the TR approach, particularly for the possibility of detecting cancers arising in the anterior zone [3,4], the “random biopsy” method is affected by a relevant incidence of missed diagnosis (10–25%) and tumor upgrading (up to 36%) in the analysis of the radical prostatectomy specimen [5]. With the advent of multiparametric magnetic resonance imaging (mpMRI), clinicians can have detailed information regarding the localization and the radiologic grade of suspicious areas within the prostate, with standardized image reporting according to the Prostate Imaging Reporting and Data System (PIRADS) score [6]. The recent literature evidence shows an increased detection of clinically significant (cs) PCa for MRI-targeted fusion biopsy (FB) compared to the ultrasound (US)-guided technique (sensitivity 0.91 vs. 0.76, respectively) [7,8], thus leading to a more precise selection of patients with the need of PCa treatment.

Superimposition of mpMRI and transrectal ultrasound (TRUS) images can either be performed mentally by the operator, who can cognitively sample the prostate through TRUS with the aid of the visual map provided by mpMRI imaging performing a visual image alignment, or with a dedicated software that records mpMRI images, matching them with the “real-time” TRUS imaging, and supports the clinician by indicating the location of the suspected area. These two techniques are known as image-guided cognitive fusion biopsy (cFB) and software-assisted fusion biopsy (saFB). 

After the introduction of MRI-based FB [9], a debate arose about the best and most accurate method for PCa detection. Indeed, each one has some advantages and disadvantages. Due to technology costs and partial availability of the systems, saFB has yet to obtain a widespread diffusion in urology departments. On the contrary, cFB relies heavily on the operator’s intuitiveness, experience, and confidence in MRI reading, without the aid of a specific software that allows them to increase the accuracy of the biopsy core collection. 

A pivotal multicentric randomized trial including 665 men with prior negative PB and a persistent clinical suspicion of PCa, demonstrated a similar detection rate of csPCa performing TR cFB, saFB, or MRI “in-bore” PB [10]. However, other reports have indicated a superiority of saFB [11]. A recent systematic review (SR) and meta-analysis (MA) including nine comparative studies on FB techniques, mostly performed with a TR approach, concluded that there was a trend toward improved rates of csPCa detection for saFB compared to cFB, although not statistically significant [12].

Accounting for the conflicting reports on the added value carried by saFB against cFB, the aim of this study is to perform a systematic review and meta-analysis of studies directly comparing csPCa detection rates for the saFB and cFB techniques, with a specific focus on their role in the TP biopsy approach. 

## 2. Materials and Methods

### 2.1. Literature Search

This SR was performed according to the 2020 Preferred Reporting Items for Systematic Reviews and Meta-Analyses (PRISMA) method [13]. 

Literature search was performed on 29th March 2023 using PubMed, EMBASE, and Scopus with no date limit. The following terms and Boolean operators were used: (cognitive fusion OR cognitive MRI OR cognitive magnetic resonance) OR (fusion OR targeted OR software-assisted OR MRI assisted OR magnetic resonance assisted) AND transperineal AND (prostate OR prostatic) AND biopsy. The review protocol was registered in PROSPERO with the registration number CRD42023418309.

### 2.2. Selection Criteria

The PICOS (Patient, Intervention, Comparison, Outcome, Study type) model was used to frame and answer the clinical question: P: Biopsy-naïve patients with clinical suspicion of prostate cancer who had a prebiopsy mpMRI of the prostate; I: Transperineal ultrasound-MRI software-assisted fusion prostate biopsy. C: Transperineal ultrasound-MRI cognitive prostate biopsy; O: primary: detection rate of clinically significant PCa (Gleason Score ≥ 3 + 4 or any core with length of cancerous tissue ≥ 4 mm) inside the targeted area; secondary: clinically insignificant PCa (Gleason Score = 3 + 3) detection rate inside the targeted area. S: retrospective, prospective, and randomized. 

### 2.3. Study Screening and Selection

Studies were accepted based on PICOS eligibility criteria. Only English papers were accepted. Animal and preclinical studies were excluded. Reviews, letters to the editor, case reports, and meeting abstracts were excluded. Studies with no data for meta-analysis were also excluded. Retrospective, prospective, and prospective randomized studies were accepted.

All retrieved studies were screened by two independent authors through Covidence systematic review software (Veritas Health Innovation, Melbourne, Australia). A third author solved discrepancies. The full text of the screened papers was selected if found pertinent to the scope of this review. 

### 2.4. Statistical Analysis

Cancer detection and complication rates were pooled using the Cochran–Mantel–Haenszel Method with the random effect model and reported as odds ratio (OR), 95% confidence interval (CI), and *p*-value. A subgroup analysis was performed for study assessing detection rates between patient and within person. Study heterogeneity was assessed utilizing the I^2^ value. Considerable heterogeneity was defined as an I^2^ value between 75% and 100%. Significance was set at *p*-value < 0.05 (two tails) and 95% CI. Meta-analysis was performed using Review Manager (RevMan) 5.4 software by Cochrane Collaboration. The quality assessment of the included studies was performed using the Cochrane Risk of Bias tool, using RoB 2 for randomized studies and ROBINS-I for retrospective and nonrandomized ones [14,15].

## 3. Results

### 3.1. Literature Screening

The literature search retrieved 1805 papers. A total of 558 duplicates were automatically excluded. Next 1247 papers were screened against title and abstract and 1234 papers were further rejected because they were unrelated to the aim of the present review. The remaining 13 full-text papers were assessed for eligibility and 5 studies were excluded. Finally, 8 papers were accepted and included [16,17,18,19,20,21,22,23]. Figure 1 shows the flow diagram of the literature search. 

### 3.2. Study Characteristics

Table 1 shows the characteristics of the included studies. There were three retrospective [16,17,20] and five prospective studies [18,19,21,22,23]. Among the latter, one study had a within-person randomization protocol [21]. There were five studies comparing the software-based versus the cognitive fusion technique between patients [16,17,18,19,20], while a comparison was performed within person in the remaining ones [21,22,23].

### 3.3. Risk of Bias Assessment

Appendix A shows the details of the quality assessment for the randomized study that showed an overall low risk of bias. Appendix A shows the details of the quality assessment for the retrospective and prospective nonrandomized studies. Overall, one study showed a serious and the remaining studies a moderate risks of bias. The most common reason for bias was due to the selection of participants, followed by bias due to confounding.

### 3.4. Meta-Analysis of Clinically Significant Prostate Cancer Detection Rates in Targeted Lesions

Meta-analysis of eight studies (1149 cases in software-based and 963 cases in cognitive fusion biopsy) showed that the detection rates of csPCa were similar between the two groups (OR 1.01, 95% CI 0.74–1.37, *p* = 0.95). Study heterogeneity was low (I^2^ 55%) (Figure 2). Subgroup analysis confirmed that there was no difference among studies comparing the two techniques between patients (OR 0.93, 95% CI 0.58–1.51, *p* = 0.78) and within person (1.17, 95% CI 0.81–1.69, *p* = 0.40). Appendix A shows a funnel plot of the meta-analysis.

### 3.5. Meta-Analysis of Clinically Insignificant Prostate Cancer Detection Rates in Targeted Lesions

Meta-analysis from six studies (458 cases in software-based and 414 cases in cognitive fusion biopsy) showed that the detection rates of clinically insignificant prostate cancer were similar between the two groups (OR 0.97, 95% CI 0.55–1.72, *p* = 0.91). Study heterogeneity was moderate (I^2^ 45%) (Figure 3). Subgroup analysis confirmed that there was no difference among studies comparing the two techniques between patients (OR 1.43, 95% CI 0.56–3.69, *p* = 0.46) and within person (OR 0.72, 95% CI 0.32–1.63, *p* = 0.43). Appendix A shows a funnel plot of the meta-analysis.

## 4. Discussion

With the publication of data from the PROMIS trial [24], mpMRI imaging of the prostate has gained a pivotal role in the diagnostic pathway of PCa, and also in biopsy-naïve patients. Indeed, it has been demonstrated that mpMRI has high sensitivity for csPCa (93%) and high negative predictive value (90–91%), thus allowing a negligible amount of significant cancers to be missed during diagnostic assessment while limiting the detection of clinically insignificant ones [25,26]. Nowadays mpMRI interpretation following the PIRADS criteria [6] represents a sort of “triage” test, leading clinicians to postpone PB in the case of negative findings and to perform an FB in the case of the detection of suspicious lesions. This trend is confirmed in all the recent reports of the literature, initially considering only men with a prior negative PB, then also at the time of the first PB (i.e., biopsy-naïve men). The multicentric randomized controlled PRECISION trial [27] demonstrated that targeted biopsies are noninferior to systematic ones, and also in biopsy-naïve patients, and confirmed that patients with a negative mpMRI can safely avoid PB. 

The subsequent question that was raised after the introduction of MRI-targeted biopsy was how to perform FB. Even if the TR route is the most popular approach among urologists for its major rapidity and tolerability in an outpatient setting, several evidences are in favor of the TP approach. In fact, the TP route seems to allow a better prostate sampling, including in the anterior zone, and is associated with a higher PCa detection rate compared to the TR one as shown in a recent multicentric study [28], with a reduced number of complications, particularly infectious [29]. Recent studies also outlined the feasibility of TP-PB without the use of prophylactic antibiotics [30,31], thus providing a further drive for the adoption of the TP approach with the purpose of minimizing the problem of antibiotic resistances. 

To the best of our knowledge, this SR and MA is the first to compare csPCa detection rates among the cFB and saFB techniques with the TP approach. The retrieved findings are mainly in line with a previous MA [12], even if most of the included studies in this paper compared PB performed with the TR route, as no significant differences in any PCa and csPCa diagnosis rates were found in the retrieved articles. As the definition of csPCa is variable among studies, we considered the most comprehensive definition of csPCa as any grade of cancer core with a length of 4 mm or greater and/or any length of cancer with a Gleason score of 3 + 4 = 7 or greater (UCL/Ahmed definition 2) [22].

Comparing the different reports, the role of operator experience appears to be crucial, particularly to perform cFB where an adequate understanding of mpMRI prostate anatomy is needed to correctly address the biopsy needle. The only discordant paper on this argument was presented by Khoo et Al [17] where the importance of operator skills appears more relevant for the saFB, where the PCa detection rate was significantly different when comparing cFb and saFB only in the subgroup of patients treated by expert operators (45.4 vs. 63.7% in the target only group and 39.4 vs. 64.5% in the target and systematic biopsy group, respectively, *p* < 0.001). As all the other reports outline that the most important advantage of saFB is to reduce the difference in inter-operator outcomes, the finding of that study may be related to the software used, which needs a rigid US-MRI fusion, thus needing more advanced operator skills.

The potential advantages of elastic over rigid fusion reconstructions remain heavily under reported to date and escape the scope of this review. While some studies seem to suggest potential accuracy advantages for elastic fusion [32,33], a plethora of commercially available platforms using a variable number of electromagnetic tracking sensors make the assessment of the added value to csPCa diagnosis between and within reconstruction types a challenging task to perform, adding another layer of complexity to the determination of the role of software image fusion in prostate biopsy [34,35]. 

As outlined by the studies included in this analysis, the operator’s confidence with MRI reading should be high in order to offer adequate cognitive biopsies. Despite this, while certification of MRI reading skills according to a standardized curriculum has been proposed for radiologists, no analogous initiative has been introduced for urologists [36]. This is especially relevant for young urologists as surveys have identified important knowledge gaps and a lack in confidence in MRI reading and interpretation worldwide [37,38]. Accounting for our results, future studies stratifying the outcomes of saFB vs. cFB per operators’ levels of experience could provide a more granular assessment of the added value of software-assisted biopsy, possibly outlining so far hidden differences between the experience groups.

Another potential advantage of saFB is the aid it provides for the identification of small volume lesions that are not always easy to locate, mainly in high-volume prostates. 

To reduce inter-patients’ variability and selection biases, the comparison of saFB with cFB was performed on the same patients in the SMART target trial [21], in which the sequence of the two strategies was also randomized within the 129 patients included. As already stated, the percentage of csPCa between saFB and cFB was comparable (54% vs. 53%, respectively, see Table 1) with only a slight increase in the median total cancer core length (6 vs. 5 mm, respectively). Valerio et al. [23] used a similar study design but without randomization of the PB sequence; in fact, each one of the 50 selected patients received saFB followed by cFB, and finally, systematic biopsy. Despite the percentage of csPCa being higher in the saFB group compared to the cFB one (51.9 vs. 44.3%), this was not statistically significant (*p* = 0.124). 

The only study where saFB was superior to cFB is the one by Patel et al. [20], in which the software-assisted fusion was performed with an elastic semi-automatic robot-assisted technique (UroBiopsyTM; Biobot Surgical). Their results may be in line with the inherent features of the robotic system, which allows for the reduction in prostate displacement and deformation during the biopsy, thus minimizing the mismatches of the fusion-guided process leading to more precise punctures and representing a promising tool to develop in the near future [39].

According to our findings, saFB and cFB can both be performed with equal effectiveness. The main limitation of this study is related to the relatively few number of included papers, which reflects a still reduced clinical adoption of TP biopsy compared to the TR one among worldwide urologists, despite the demonstrated advantages of the TP approach in terms of the PCa detection rate and reduced bleeding and infectious complications [40]. Other potential limitations of this MA are related to the different software, different learning curves and operator experiences, and different clinical indications between each study. We decided not to report the data from systematic biopsies performed in the selected studies, as SBs were not objective in this study. Moreover, a recent paper by Porpiglia et al. [41] demonstrated that FB alone is not inferior to FB + SB in the detection of csPCa, so SB does not appear to be mandatory in this setting.

An adequate knowledge of prostatic mpMRI appears to be mandatory for the clinician performing the biopsy, despite elastic MRI fusion software providing some support to target identification during the image fusion procedure. 

## 5. Conclusions

According to our findings, there is no actual evidence for a better method to perform TRUS-MRI transperineal fusion biopsy. Clinician experience and technologic availability appear to be the main determinants in the choice of one technique over the other. Technological advances and an increased understanding and standardization of mpMRI imaging will further drive improvements toward the implementation, feasibility, and efficacy of TP prostate biopsy techniques. Future assessment of the role of the operator’s experience and different reconstruction techniques for software fusion may provide a more comprehensive understanding of the role of TRUS-MRI software fusion.

## Figures and Tables

**Figure 1 cancers-15-03443-f001:**
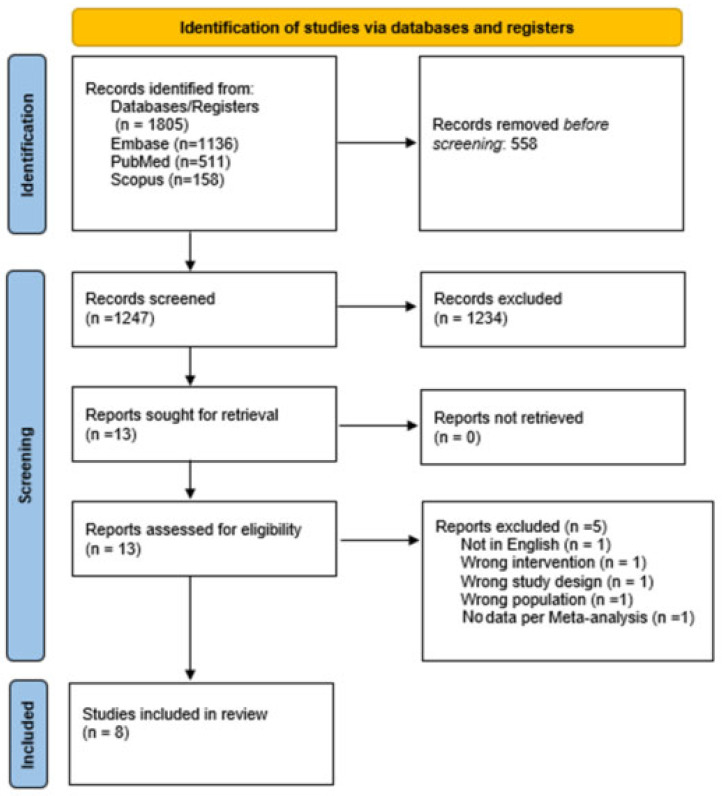
PRISMA study flowchart [13].

**Figure 2 cancers-15-03443-f002:**
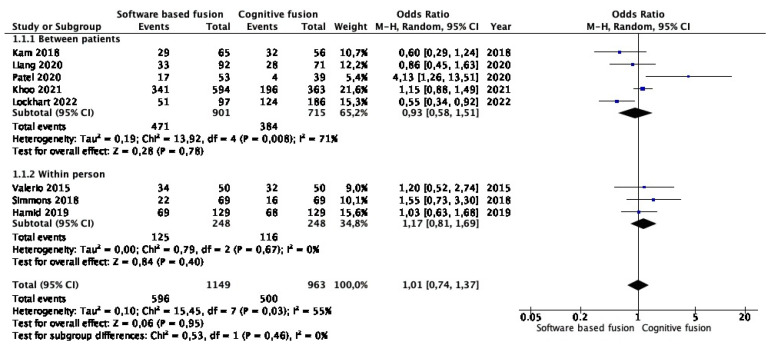
Meta-analysis of clinically significant prostate cancer detection rates in targeted lesions [16,17,18,19,20,21,22,23].

**Figure 3 cancers-15-03443-f003:**
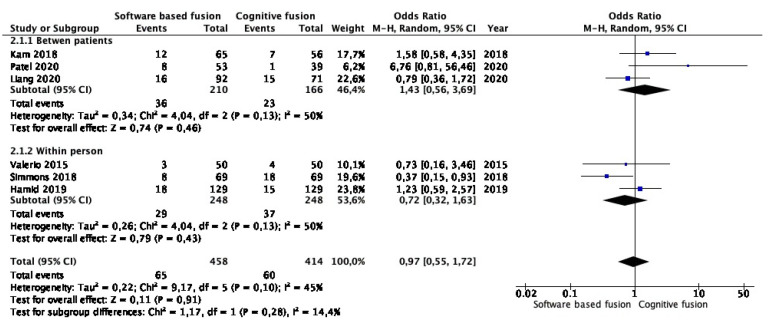
Meta-analysis of clinically insignificant prostate cancer detection rates in targeted lesions [16,18,20,21,22,23].

**Table 1 cancers-15-03443-t001:** Study characteristics.

Authors [REF]	Study Design	Fusion Type	saFB Cohort						cFB Cohort						Final Comment
			Patients *n*	PSA ng/mL *(%)*	MRI Targets *n (%)*	Targeted Cores *n*	csPCa Target Biopsy *n (%)*	ciPCa Target Biopsy *n (%)*	Patients *n*	PSA ng/mL *(%)*	MRI Targets *n (%)*	Targeted Cores *n*	csPCa Target Biopsy *n (%)*	ciPCa Target Biopsy *n (%)*	
Liang et al. 2020 [16]	Retrospective	Rigid (Predictive Fusion Software; BK medical, Herlev, Denmark)	92	8.03	NR	4	33 (35.87%)	14 (15.2%)	71	7.66	NR	4	28 (39.43)	15 (21.1%)	Cognitive and fusion targeting detect similar rates of csPCa
Khoo et al. 2021 [17]	Retrospective	Elastic (BiopSee, Medcom, Darmstadt, Germany	594	7.9	NR	NR	341 (57.4%)	NR	363	8.4	NR	NR	196 (54.0%)		Cognitive and fusion targeting detect similar rates of csPCa, although fusion biopsy may be superior in experienced hands
Kam et al. 2018 [18]	Prospective	Not specified (Biojet, D&K Technologies GmbH, Barum, Germany	65	7.3	PIRADS 3: 28 (43%); PIRADS4 or 5:37 (57%)	4.6	29 (44.6%)	12 (18.4%)	56	7.5	PIRADS 3: 18 (32%) PIRADS 4 or 5: 38 (68%)	3.1	32 (57.1%)	7 (12.5%)	Cognitive and fusion targeting detect similar rates of csPCa. Cognitive biopsy had a significantly higher core positivity rate than fusion biopsy.
Lockhart et al. 2022 [19]	Prospective	Not specified (MIM Bx, MIM Software Inc, Cleveland, OH, USA	131	5.8	NR	NR	52 (39.70%)	NR	224	7.64	NR	NR	120 (53.60%)	NR	Cognitive and fusion targeting detect similar rates of csPCa
Patel et al. 2020 [20]	Retrospective	Elastic (Urofusion, Biobot Surgical, Singapore, Singapore)	53	<4: 11 (20.8%) 4-10: 40 (75.5%) >10: 2 (3.8%)	PIRADS 3: 14 (26.4%) PIRADS 4: 28 (58.2%) PIRADS 5:11 (20.8%)	4	17 (32.1%)	8 (15.1%)	39	<4: 9 (23.1%) 4-10: 22 (56.4%) >10: 8 (20.5%)	PIRADS 3: 14 (38.5%) PIRADS 4: 13 (33.3%) PIRADS 5: 11 (28.2%)	3	4 (10.3%)	1 (2.6%)	Robot-assisted fusion targeting detects a significantly higher percentage of csPCa than cognitive targeting
Hamid et al. 2019 [21]	Randomized Controlled Trial	Elastic (SmartTarget software, London, UK)	129	8.5	Likert 3: 22 (17%) Likert 4: 67 (52%) Likert 5: 40 (31%)	3	69 (54%)	18 (14%)	129	8.5	Likert 3: 22 (17%) Likert 4: 67 (52%) Likert 5: 40 (31%)	3	68 (53%)	15 (12%)	Cognitive and fusion targeting detect similar rates of csPCa
Simmons et al. 2018 [22]	Prospective, Comparative Trial	Elastic (SmartTarget software, London, UK)	69	NR	Likert	4	22 (31.8%)	8 (11.6%)	69	NR	Likert	4	16 (23.2%)	18 (26.1%)	csPCa defined as any grade of cancer core with a length of 4 mm or greater and/or any length of cancer with a Gleason score of 3 + 4 = 7 or greater (UCL/Ahmed definition)—Cognitive and fusion targeting detect similar rates of csPCa
Valerio et al. 2015 [23]	Prospective	Rigid (Biojet, D&K Technologies GmbH, Barum, Germany)	50	7.9	Likert 3: 27 (34%) Likert 4: 28 (35%) Likert 5: 24 (31%)	3	34 (68%)	3 (6%)	50	7.9	Likert 3: 27 (34%) Likert 4: 28 (35%) Likert 5: 24 (31%)	4	32 (64%)	4 (8%)	Cognitive and fusion targeting detect similar rates of csPCa

## Data Availability

Data are available in MEDLINE, PubMed, and Scopus. Full references are available in the text.

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
