# Peer review of "Transperineal US-MRI Fusion-Guided Biopsy for the Detection of Clinical Significant Prostate Cancer: A Systematic Review and Meta-Analysis Comparing Cognitive and Software-Assisted Technique"

_cancers, 2023, doi:10.3390/cancers15133443_

Round 1
Reviewer 1 Report
The paper is really interesting and well-conducted. It is a meta-analysis about two different ways to perform prostatic biopsy. Only 8 study are involved in the review but they are homogeneous for PSA, targeted cores and results. Probably it should be fine to have more prospective or randomized studies in the future.
Author Response
We thank Reviewer 1 for the appreciation of Our work. US-MRI fusion biopsy represents the future gold standard of prostate biopsy, therefore further comparative papers are needed in the next future to assess for potential differences in PCa detection rate. The principal aim of this paper is to give an overview of the actual clinical scenario and to promote further researches on this field.
Reviewer 2 Report
Dear Authors, your manuscript is interesting and on a current topic.
The research strategy and methodology are good. Please add the AMSTAR scale.
The definition of csPca is different among the different papers included in your MA and it could be affected your findings. Can you explain how you opted to manage this aspect? Is possibile to standardize this definition before to proceed to MA?
The paper that you included did only TB or also SB?
Your data and results are affected by several bias such as different softwares, different learning curve, maybe different indications....please improve the Limitation section into the discussion
The following paper could be cited into the discussion:
-- A prospective randomized controlled trial comparing target prostate biopsy alone approach vs. target plus standard in naïve patients with positive mpMRI. Minerva Urol Nephrol. 2023 Feb;75(1):31-41. doi: 10.23736/S2724-6051.22.05189-8. Epub 2023 Jan 10. PMID: 36626117.
Author Response
Dear reviewer, thanks for Your kind comments.
Regarding the requested AMSTAR scale, we already provided all the requested points of the AMSTAR scale in the Methods section. The rating of overall confidence in the study results is High, without any critical weakness. AMSTAR scale is a tool for assessing the methodological quality of the SR, but we do not believe that this Is mandatory in this case, as the Methods section already assess the high quality of research methodology and of included study. However, we are ready to add the reference of AMSTAR scale and the AMSTAR flowchart in the paper if You feel this necessary.
As the definition of csPCa varied between different studies, as reported in the Discussion section, we considered the most comprehensive definition of cs PCa as any grade of cancer core with a length of 4 mm or greater and/ or any length of cancer with a Gleason score of 3 + 4= 7 or greater (UCL/Ahmed definition 2). The result of all different studies were reported and standardized following this definition in this MA, data can be read on the Study Table. The included papers performed also SB, but those were not object of this study, so we focused on target biopsy only.
We improved study limitations according to Your commentaries (see line 318-321).
We thank specially You for the last suggestion, we included the reference of the article by Porpiglia et Al [REF 40], which gives further strength to our findings, showing that SB is not mandatory as does not improve csPCa detection rate.
Reviewer 3 Report
This manuscript presents a meta-analysis in comparing the detection rate of clinically significant PCa between the cognitive and software-assisted MRI-US Fusion Biopsy technique.
The authors emphasize the advantages of transperineal prostate biopsies over transrectal biopsies. However, by far the majority of practicing urologists perform the transrectal approach. Therefore, the authors should provide with a reference the percentage rate for the transperineal approach (which is low).
Line 85-87: It should be mentioned that the recent systemic review and meta-analysis (reference # 12) was done on transrectal MRI-US Fusion biopsies, and not on perineal MRI-Fusion biopsies (as in this review study).
At the end of the Discussion section the authors should more elaborate on the limitations of this review report.
Author Response
Dear Reviewer, thanks for Your appreciation for this paper. As You correctly say, the transrectal approach is still the most performed from urologists worldwide. In fact, most of comparative studies between cognitive and software assisted PB consider the transrectal approach (as reported in a previous MA on this argument, cited in REF 12). However, TP biopsy harbors relevant advantages despite a longer operator learning curve and a slight higher patient discomfort, as detailed reported in a previous paper from Our working group, which we now included in references (Martorana E, Pirola GM, Aisa MC, Scialpi P, Di Blasi A, Saredi G, D'Andrea A, Signore S, Grisanti R, Scialpi M. Prostate MRI and transperineal TRUS/MRI fusion biopsy for prostate cancer detection: clinical practice updates. Turk J Urol. 2019 Jul 1;45(4):237-244. doi: 10.5152/tud.2019.19106. PMID: 31291186; PMCID: PMC6619838.).
According to Your correct comment, we emphasized this concept among study limitations.